# Spectral Embedding of Regularized Block Models

**Nathan De Lara & Thomas Bonald**
Institut Polytechnique de Paris
Paris, France
{nathan.delara, thomas.bonald}@telecom-paris.fr

## Abstract

Spectral embedding is a popular technique for the representation of graph data. Several regularization techniques have been proposed to improve the quality of the embedding with respect to downstream tasks like clustering. In this paper, we explain on a simple block model the impact of the complete graph regularization, whereby a constant is added to all entries of the adjacency matrix. Specifically, we show that the regularization forces the spectral embedding to focus on the largest blocks, making the representation less sensitive to noise or outliers. We illustrate these results on both on both synthetic and real data, showing how regularization improves standard clustering scores.

## 1 Introduction

Spectral embedding is a standard technique for the representation of graph data (Ng et al., 2002; Belkin & Niyogi, 2002). Given the adjacency matrix $A \in \mathbb{R}_+^{n \times n}$ of the graph, it is obtained by solving either the eigenvalue problem:

$$LX = X\Lambda, \text{ with } X^T X = I, \tag{1}$$

or the generalized eigenvalue problem:

$$LX = DX\Lambda, \text{ with } X^T DX = I, \tag{2}$$

where $D = \text{diag}(A1_n)$ is the degree matrix, with $1_n$ the all-ones vector of dimension $n$, $L = D - A$ is the Laplacian matrix of the graph, $\Lambda \in \mathbb{R}^{k \times k}$ is the diagonal matrix of the $k$ smallest (generalized) eigenvalues of $L$ and $X \in \mathbb{R}^{n \times k}$ is the corresponding matrix of (generalized) eigenvectors. In this paper, we only consider the generalized eigenvalue problem, whose solution is given by the spectral decomposition of the normalized Laplacian matrix $L_{\text{norm}} = I - D^{-1/2}AD^{-1/2}$ (Luxburg, 2007).

The spectral embedding can be interpreted as equilibrium states of some physical systems (Snell & Doyle, 2000; Spielman, 2007; Bonald et al., 2018), a desirable property in modern machine learning. However, it tends to produce poor results on real datasets if applied directly on the graph (Amini et al., 2013). One reason is that real graphs are most often disconnected due to noise or outliers in the dataset.

In order to improve the quality of the embedding, two main types of regularization have been proposed. The first artificially increases the degree of each node by a constant factor (Chaudhuri et al., 2012; Qin & Rohe, 2013), while the second adds a constant to all entries of the original adjacency matrix (Amini et al., 2013; Joseph et al., 2016; Zhang & Rohe, 2018). In the practically interesting case where the original adjacency matrix $A$ is sparse, the regularized adjacency matrix is dense but has a so-called sparse + low rank structure, enabling the computation of the spectral embedding on very large graphs (Lara, 2019).

While (Zhang & Rohe, 2018) explains the effects of regularization through graph conductance and (Joseph et al., 2016) through eigenvector perturbation on the Stochastic Block Model, there is no simple interpretation of the benefits of graph regularization. In this paper, we show on a simple block model that the complete graph regularization forces the spectral embedding to separate the blocks in decreasing order of size, making the embedding less sensitive to noise or outliers in the data.

Indeed, (Zhang & Rohe, 2018) identified that, without regularization, the cuts corresponding to the first dimensions of the spectral embedding tend to separate small sets of nodes, so-called dangling sets, loosely connected to the rest of the graph. Our work shows more explicitly that regularization forces the spectral embedding to focus on the largest clusters. Moreover, our analysis involves some explicit characterization of the eigenvalues, allowing us to quantify the impact of the regularization parameter.

The rest of this paper is organized as follows. Section 2 presents block models and an important preliminary result about their aggregation. Section 3 presents the main result of the paper, about the regularization of block models, while Section 4 extends this result to bipartite graphs. Section 5 presents the experiments and Section 6 concludes the paper.

## 2 AGGREGATION OF BLOCK MODELS

Let $A \in \mathbb{R}_+^{n \times n}$ be the adjacency matrix of an undirected, weight graph, that is a symmetric matrix such that $A_{ij} > 0$ if and only if there is an edge between nodes $i$ and $j$, with weight $A_{ij}$. Assume that the $n$ nodes of the graph can be partitioned into $K$ blocks of respective sizes $n_1, \ldots, n_K$ so that any two nodes of the same block have the same neighborhood, i.e., the corresponding rows (or columns) of $A$ are the same. Without any loss of generality, we assume that the matrix $A$ has rank $K$. We refer to such a graph as a block model.

Let $Z \in \mathbb{R}^{n \times K}$ be the associated membership matrix, with $Z_{ij} = 1$ if index $i$ belongs to block $j$ and 0 otherwise. We denote by $W = Z^T Z \in \mathbb{R}^{K \times K}$ the diagonal matrix of block sizes.

Now define $\bar{A} = Z^T A Z \in \mathbb{R}^{K \times K}$. This is the adjacency matrix of the aggregate graph, where each block of the initial graph is replaced by a single node; two nodes in this graph are connected by an edge of weight equal to the total weight of edges between the corresponding blocks in the original graph. We denote by $\bar{D} = \text{diag}(\bar{A} 1_K)$ the degree matrix and by $\bar{L} = \bar{D} - \bar{A}$ the Laplacian matrix of the aggregate graph.

The following result shows that the solution to the generalized eigenvalue problem (2) follows from that of the aggregate graph:

**Proposition 1.** *Let $x$ be a solution to the generalized eigenvalue problem:*

$$Lx = \lambda Dx. \tag{3}$$

*Then either $Z^T x = 0$ and $\lambda = 1$ or $x = Zy$ where $y$ is a solution to the generalized eigenvalue problem:*

$$\bar{L}y = \lambda \bar{D}y. \tag{4}$$

*Proof.* Consider the following reformulation of the generalized eigenvalue problem (3):

$$Ax = Dx(1 - \lambda). \tag{5}$$

Since the rank of $A$ is equal to $K$, there are $n - K$ eigenvectors $x$ associated with the eigenvalue $\lambda = 1$, each satisfying $Z^T x = 0$. By orthogonality, the other eigenvectors satisfy $x = Zy$ for some vector $y \in \mathbb{R}^K$. We get:

$$AZy = DZy(1 - \lambda),$$

so that

$$\bar{A}y = \bar{D}y(1 - \lambda).$$

Thus $y$ is a solution to the generalized eigenvalue problem (4). $\square$

## 3 REGULARIZATION OF BLOCK MODELS

Let $A$ be the adjacency matrix of some undirected graph. We consider a regularized version of the graph where an edge of weight $\alpha$ is added between all pairs of nodes, for some constant $\alpha > 0$. The corresponding adjacency matrix is given by:

$$A_\alpha = A + \alpha J,$$

where $J = 1_n 1_n^T$ is the all-ones matrix of same dimension as $A$. We denote by $D_\alpha = \text{diag}(A_\alpha 1_n)$ the corresponding degree matrix and by $L_\alpha = D_\alpha - A_\alpha$ the Laplacian matrix.

We first consider a simple block model where the graph consists of $K$ disjoint cliques of respective sizes $n_1 > n_2 > \cdots > n_K$ nodes, with $n_K \geq 1$. In this case, we have $A = ZZ^T$, where $Z$ is the membership matrix.

The objective of this section is to demonstrate that, in this setting, the $k$-th dimension of the spectral embedding isolates the $k-1$ largest cliques from the rest of the graph, for any $k \in \{2, \ldots, K\}$

**Lemma 1.** *Let $\lambda_1 \leq \lambda_2 \leq \ldots \leq \lambda_n$ be the eigenvalues associated with the generalized eigenvalue problem:*

$$L_\alpha x = \lambda D_\alpha x. \tag{6}$$

*We have $\lambda_1 = 0 < \lambda_2 \leq \ldots \leq \lambda_K < \lambda_{K+1} = \ldots = \lambda_n = 1$.*

*Proof.* Since the Laplacian matrix $L_\alpha$ is positive semi-definite, all eigenvalues are non-negative (Chung, 1997). We know that the eigenvalue 0 has multiplicity 1 on observing that the regularized graph is connected. Now for any vector $x$,

$$x^T A_\alpha x = x^T A x + \alpha x^T J x = ||Z^T x||^2 + \alpha (1_n^T x)^2 \geq 0,$$

so that the matrix $A_\alpha$ is positive semi-definite. In view of (5), this shows that $\lambda \leq 1$ for any eigenvalue $\lambda$. The proof then follows from Proposition 1, on observing that the eigenvalue 1 has multiplicity $n - K$. $\square$

**Lemma 2.** *Let $x$ be a solution to the generalized eigenvalue problem (6) with $\lambda \in (0, 1)$. There exists some $s \in \{+1, -1\}$ such that for each node $i$ in block $j$,*

$$\text{sign}(x_i) = s \iff n_j \geq \alpha \frac{1-\lambda}{\lambda} n.$$

*Proof.* In view of Proposition 1, we have $x = Zy$ where $y$ is a solution to the generalized eigenvalue problem of the aggregate graph, with adjacency matrix:

$$\bar{A}_\alpha = Z^T A_\alpha Z = Z^T (A + \alpha J) Z.$$

Since $A = ZZ^T$ and $W = Z^T Z$, we have $\bar{A}_\alpha = W^2 + \alpha Z^T J Z$. Using the fact that $Z1_K = 1_n$, we get $J = 1_n 1_n^T = ZJ_K Z^T$ with $J_K = 1_K 1_K^T$ the all-ones matrix of dimension $K \times K$, so that:

$$\bar{A}_\alpha = W(I_K + \alpha J_K)W,$$

where $I_K$ is the identity matrix of dimension $K \times K$. We deduce the degree matrix:

$$\bar{D}_\alpha = W(W + \alpha n I_K),$$

and the Laplacian matrix:

$$\bar{L}_\alpha = \bar{D}_\alpha - \bar{A}_\alpha = \alpha W(nI_K - J_K W).$$

The generalized eigenvalue problem associated with the aggregate graph is:

$$\bar{L}_\alpha y = \lambda \bar{D}_\alpha y.$$

After multiplication by $W^{-1}$, we get:

$$\alpha(nI_K - J_K W)y = \lambda(W + \alpha n I_K)y.$$

Observing that $J_K W y = 1_K 1_K^T W y = (1_K^T W y) 1_K \propto 1_K$, we conclude that:

$$(\alpha n(1-\lambda) - \lambda W)y \propto 1_K, \tag{7}$$

and since $W = \text{diag}(n_1, \ldots, n_K)$,

$$\forall j = 1, \ldots, K, \quad y_j \propto \frac{1}{\lambda n_j - \alpha(1-\lambda)n}. \tag{8}$$

The result then follows from the fact that $x = Zy$. $\square$

**Lemma 3.** *The $K$ smallest eigenvalues satisfy:*

$$0 = \lambda_1 < \mu_1 < \lambda_2 < \mu_2 < \cdots < \lambda_K < \mu_K,$$

*where for all $j = 1, \ldots, K$,*

$$\mu_j = \frac{\alpha n}{\alpha n + n_j}.$$

*Proof.* We know from Lemma 1 that the $K$ smallest eigenvalues are in $[0, 1)$. Let $x$ be a solution to the generalized eigenvalue problem (6) with $\lambda \in (0, 1)$. We know that $x = Zy$ where $y$ is an eigenvector associated with the same eigenvalue $\lambda$ for the aggregate graph. Since $1_K$ is an eigenvector for the eigenvalue 0, we have $y^T \bar{D}_\alpha 1_K = 0$. Using the fact that $\bar{D}_\alpha = W(W + \alpha n I_K)$, we get:

$$\sum_{j=1}^{K} n_j (n_j + \alpha n) y_j = 0.$$

We then deduce from (7) and (8) that $\lambda \notin \{\mu_1, \ldots, \mu_K\}$ and

$$\sum_{j=1}^{K} n_j (n_j + \alpha n) \frac{1}{\lambda/\mu_j - 1} = 0.$$

This condition cannot be satisfied if $\lambda < \mu_1$ or $\lambda > \mu_K$ as the terms of the sum would be either all positive or all negative.

Now let $y'$ be another eigenvector for the aggregate graph, with $y^T \bar{D}_\alpha y' = 0$, for the eigenvalue $\lambda' \in (0, 1)$. By the same argument, we get:

$$\sum_{j=1}^{K} n_j (n_j + \alpha n) y_j y'_j = 0,$$

and

$$\sum_{j=1}^{K} n_j (n_j + \alpha n) \frac{1}{\lambda/\mu_j - 1} \frac{1}{\lambda'/\mu_j - 1} = 0.$$

with $\lambda' \notin \{\mu_1, \ldots, \mu_K\}$. This condition cannot be satisfied if $\lambda$ and $\lambda'$ are in the same interval $(\mu_j, \mu_{j+1})$ for some $j$ as the terms in the sum would be all positive. There are $K - 1$ eigenvalues in $(0, 1)$ for $K - 1$ such intervals, that is one eigenvalue per interval. □

The main result of the paper is the following, showing that the $k - 1$ largest cliques of the original graph can be recovered from the spectral embedding of the regularized graph in dimension $k$.

**Theorem 1.** *Let $X$ be the spectral embedding of dimension $k$, as defined by (2), for some $k$ in the set $\{2, \ldots, K\}$. Then $\text{sign}(X)$ gives the $k - 1$ largest blocks of the graph.*

*Proof.* Let $x$ be the $j$-th column of the matrix $X$, for some $j \in \{2, \ldots, k\}$. In view of Lemma 3, this is the eigenvector associated with eigenvalue $\lambda_j \in (\mu_{j-1}, \mu_j)$, so that

$$\alpha \frac{1 - \lambda_j}{\lambda_j} n \in (n_{j-1}, n_j).$$

In view of Lemma 2, all entries of $x$ corresponding to blocks of size $n_1, n_2 \ldots, n_{j-1}$ have the same sign, the other having the opposite sign. □

Theorem 1 can be extended in several ways. First, the assumption of distinct block sizes can easily be relaxed. If there are $L$ distinct values of block sizes, say $m_1, \ldots, m_L$ blocks of sizes $n_1 > \ldots > n_L$, there are $L$ distinct values for the thresholds $\mu_j$ and thus $L$ distinct values for the eigenvalues $\lambda_j$ in $[0, 1)$, the multiplicity of the $j$-th smallest eigenvalue being equal to $m_j$. The spectral embedding in dimension $k$ still gives $k - 1$ cliques of the largest sizes.

Second, the graph may have edges between blocks. Taking $A = ZZ^T + \varepsilon J$ for instance, for some parameter $\varepsilon \geq 0$, the results are exactly the same, with $\alpha$ replaced by $\epsilon + \alpha$. A key observation is that regularization really matters when $\varepsilon \to 0$, in which case the initial graph becomes disconnected and, in the absence of regularization, the spectral embedding may isolate small connected components of the graph. In particular, the regularization makes the spectral embedding much less sensitive to noise, as will be demonstrated in the experiments.

Finally, degree correction can be added by varying the node degrees within blocks. Taking $A = \theta Z Z^T \theta$, for some arbitrary diagonal matrix $\theta$ with positive entries, similar results can be obtained under the regularization $A_\alpha = A + \alpha \theta J \theta$. Interestingly, the spectral embedding in dimension $k$ then recovers the $k-1$ largest blocks in terms of *normalized weight*, the ratio of the total weight of the block to the number of nodes in the block.

## 4 REGULARIZATION OF BIPARTITE GRAPHS

Let $B = \mathbb{R}_+^{n \times m}$ be the biadjacency matrix of some bipartite graph with respectively $n, m$ nodes in each part, i.e., $B_{ij} > 0$ if and only if there is an edge between node $i$ in the first part of the graph and node $j$ in the second part of the graph, with weight $B_{ij}$. This is an undirected graph of $n + m$ nodes with adjacency matrix:

$$A = \begin{bmatrix} 0 & B \\ B^T & 0 \end{bmatrix}$$

The spectral embedding of the graph (2) can be written in terms of the biadjacency matrix as follows:

$$\begin{cases} B X_2 = D_1 X_1 (I - \Lambda) \\ B^T X_1 = D_2 X_2 (I - \Lambda) \end{cases} \tag{9}$$

where $X_1, X_2$ are the embeddings of each part of the graph, with respective dimensions $n \times k$ and $m \times k$, $D_1 = \text{diag}(B 1_m)$ and $D_2 = \text{diag}(B^T 1_n)$. In particular, the spectral embedding of the graph follows from the generalized SVD of the biadjacency matrix $B$.

The complete regularization adds edges between all pairs of nodes, breaking the bipartite structure of the graph. Another approach consists in applying the regularization to the biadjacency matrix, i.e., in considering the regularized bipartite graph with biadjacency matrix:

$$B_\alpha = B + \alpha J,$$

where $J = 1_n 1_m^T$ is here the all-ones matrix of same dimension as $B$. The spectral embedding of the regularized graph is that associated with the adjacency matrix:

$$A_\alpha = \begin{bmatrix} 0 & B_\alpha \\ B_\alpha^T & 0 \end{bmatrix} \tag{10}$$

As in Section 3, we consider a block model so that the biadjacency matrix $B$ is block-diagonal with all-ones block matrices on the diagonal. Each part of the graph consists of $K$ groups of nodes of respective sizes $n_1 > \ldots > n_K$ and $m_1 > \ldots > m_K$, with nodes of block $j$ in the first part connected only to nodes of block $j$ in the second part, for all $j = 1, \ldots, K$.

We consider the generalized eigenvalue problem (6) associated with the above matrix $A_\alpha$. In view of (9), this is equivalent to the generalized SVD of the regularized biadjacency matrix $B_\alpha$. We have the following results, whose proofs are deferred to the appendix:

**Lemma 4.** *Let $\lambda_1 \leq \lambda_2 \leq \ldots \leq \lambda_n$ be the eigenvalues associated with the generalized eigenvalue problem (6). We have $\lambda_1 = 0 < \lambda_2 \leq \ldots \leq \lambda_K < \lambda_{K+1} = \ldots = \lambda_{n-2K} < \ldots < \lambda_n = 2$.*

**Lemma 5.** *Let $x$ be a solution to the generalized eigenvalue problem (6) with $\lambda \in (0, 1)$. There exists $s_1, s_2 \in \{+1, -1\}$ such that for each node $i$ in block $j$ of part $p \in \{1, 2\}$,*

$$\text{sign}(x_i) = s_p \quad \Longleftrightarrow \quad \frac{n_j m_j}{(n_j + \alpha n)(m_j + \alpha m)} \geq 1 - \lambda.$$

**Lemma 6.** *The $K$ smallest eigenvalues satisfy:*

$$0 = \lambda_1 < \mu_1 < \lambda_2 < \mu_2 < \cdots < \lambda_K < \mu_K,$$

*where for all $j = 1, \ldots, K$,*

$$\mu_j = 1 - \frac{n_j m_j}{(n_j + \alpha n)(m_j + \alpha m)}.$$

**Theorem 2.** *Let $X$ be the spectral embedding of dimension $k$, as defined by (2), for some $k$ in the set $\{2, \ldots, K\}$. Then $\text{sign}(X)$ gives the $k-1$ largest blocks of each part of the graph.*

Like Theorem 1, the assumption of decreasing block sizes can easily be relaxed. Assume that block pairs are indexed in decreasing order of $\mu_j$. Then the spectral embedding of dimension $k$ gives the $k-1$ first block pairs for that order. It is interesting to notice that the order now depends on $\alpha$: when $\alpha \to 0^+$, the block pairs $j$ of highest value $(\frac{n}{n_j} + \frac{m}{m_j})^{-1}$ (equivalently, highest *harmonic mean* of proportions of nodes in each part of the graph) are isolated first; when $\alpha \to +\infty$, the block pairs $j$ of highest value $\frac{n_j m_j}{nm}$ (equivalently, the highest *geometric mean* of proportions of nodes in each part of the graph) are isolated first.

The results also extend to non-block diagonal biadjacency matrices $B$ and degree-corrected models, as for Theorem 1.

## 5 EXPERIMENTS

We now illustrate the impact of regularization on the quality of spectral embedding. We focus on a clustering task, using both synthetic and real datasets where the ground-truth clusters are known. In all experiments, we skip the first dimension of the spectral embedding as it is not informative (the corresponding eigenvector is the all-ones vector, up to some multiplicative constant). The code to reproduce these experiments is available online[1].

### 5.1 TOY GRAPH

We first illustrate the theoretical results of the paper with a toy graph consisting of 3 cliques of respective sizes $5, 3, 2$. We compute the spectral embeddings in dimension 1, using the second smallest eigenvalue. Denoting by $Z$ the membership matrix, we get $X \approx Z(-0.08, 0.11, 0.05)^T$ for $\alpha = 1$, showing that the embedding isolates the largest cluster; this is not the case in the absence of regularization, where $X \approx Z(0.1, -0.1, 0.41)^T$.

### 5.2 DATASETS

This section describes the datasets used in our experiments. All graphs are considered as undirected. Table 1 presents the main features of the graphs.

**Stochastic Block-Model (SBM)**   We generate 100 instances of the same stochastic block model (Holland et al., 1983). There are 100 blocks of size 20, with intra-block edge probability set to 0.5 for the first 50 blocks and 0.05 for the other blocks. The inter-block edge probability is set to 0.001 Other sets of parameters can be tested using the code available online. The ground-truth cluster of each node corresponds to its block.

**20newsgroup (NG)**   This dataset consists of around 18000 newsgroups posts on 20 topics. This defines a weighted bipartite graph between documents and words. The label of each document corresponds to the topic.

**Wikipedia for Schools (WS)**   (Haruechaiyasak & Damrongrat, 2008). This is the graph of hyperlinks between a subset of Wikipedia pages. The label of each page is its category (e.g., countries, mammals, physics).

Table 1: Main features of the graphs.

| dataset | SBM | NG | WS |
|---|---|---|---|
| # nodes ($n$) | 2000 | 10723 | 4591 |
| # edges | $\approx 5.10^3$ | $\approx 2.10^6$ | $\approx 2.10^5$ |
| # clusters in ground truth | 100 | 20 | 14 |

---

[1]`https://github.com/nathandelara/Spectral-Embedding-of-Regularized-Block-Models/`

## 5.3 METRICS

We consider a large set of metrics from the clustering literature. All metrics are upper-bounded by 1 and the higher the score the better.

**Homogeneity (H), Completeness (C) and V-measure score (V)**   (Rosenberg & Hirschberg, 2007). Supervised metrics. A cluster is homogeneous if all its data points are members of a single class in the ground truth. A clustering is complete if all the members of a class in the ground truth belong to the same cluster in the prediction. Harmonic mean of homogeneity and completeness.

**Adjusted Rand Index (ARI)**   (Hubert & Arabie, 1985). Supervised metric. This is the corrected for chance version of the Rand Index which is itself an accuracy on pairs of samples.

**Adjusted Mutual Information (AMI)**   (Vinh et al., 2010) Supervised metric. Adjusted for chance version of the mutual information.

**Fowlkes-Mallows Index (FMI)**   (Fowlkes & Mallows, 1983). Supervised metric. Geometric mean between precision and recall on the edge classification task, as described for the ARI.

**Modularity (Q)**   (Newman, 2006). Unsupervised metric. Fraction of edges within clusters compared to that is some null model where edges are shuffled at random.

**Normalized Standard Deviation (NSD)**   Unsupervised metric. 1 minus normalized standard deviation in cluster size.

## 5.4 EXPERIMENTAL SETUP

All graphs are embedded in dimension 20, with different regularization parameters. To compare the impact of this parameter across different datasets, we use a relative regularization parameter $(w/n^2)\alpha$, where $w = 1_n^T A 1_n$ is the total weight of the graph.

We use the K-Means algorithm with to cluster the nodes in the embedding space. The parameter $K$ is set to the ground-truth number of clusters (other experiments with different values of $K$ are reported in the Appendix). We use the Scikit-learn (Pedregosa et al., 2011) implementation of K-Means and the metrics, when available. The spectral embedding and the modularity are computed with the Scikit-network package, see the documentation for more details[2].

## 5.5 RESULTS

We report the results in Table 2 for relative regularization parameter $\alpha = 0, 0.1, 1, 10$. We see that the regularization generally improves performance, the optimal value of $\alpha$ depending on both the dataset and the score function. As suggested by Lemma 3, the optimal value of the regularization parameter should depend on the distribution of cluster sizes, on which we do not have any prior knowledge.

To test the impact of noise on the spectral embedding, we add isolated nodes with self loop to the graph and compare the clustering performance with and without regularization. The number of isolated nodes is given as a fraction of the initial number of nodes in the graph. Scores are computed only on the initial nodes. The results are reported in Table 3 for the Wikipedia for Schools dataset. We observe that, in the absence of regularization, the scores drop even with only 1% noise. The computed clustering is a trivial partition with all initial nodes in the same cluster. This means that the 20 first dimensions of the spectral embedding focus on the isolated nodes. On the other hand, the scores remain approximately constant in the regularized case, which suggests that regularization makes the embedding robust to this type of noise.

---

[2]https://scikit-network.readthedocs.io/

Table 2: Impact of regularization on clustering performance.

SBM

| $\alpha$ | H | C | V | ARI | AMI | FMI | Q | NSD |
|---|---|---|---|---|---|---|---|---|
| 0 | 0.19 | 0.27 | 0.22 | 0.0 | **0.01** | **0.03** | 0.45 | 0.76 |
| 0.1 | 0.33 | 0.35 | 0.34 | 0.0 | **0.01** | 0.01 | **0.52** | 0.91 |
| 1 | **0.36** | **0.37** | **0.36** | 0.0 | **0.01** | 0.01 | 0.50 | **0.92** |
| 10 | 0.28 | 0.34 | 0.30 | 0.0 | 0.00 | 0.02 | 0.36 | 0.78 |

NG

| $\alpha$ | H | C | V | ARI | AMI | FMI | Q | NSD |
|---|---|---|---|---|---|---|---|---|
| 0 | 0.40 | **0.70** | 0.51 | 0.19 | 0.50 | 0.34 | **0.21** | 0.55 |
| 0.1 | 0.44 | **0.70** | **0.54** | **0.22** | **0.54** | **0.35** | **0.21** | 0.59 |
| 1 | **0.46** | 0.67 | **0.54** | 0.20 | **0.54** | 0.33 | 0.20 | **0.60** |
| 10 | 0.37 | 0.55 | 0.45 | 0.13 | 0.44 | 0.26 | 0.17 | 0.56 |

WS

| $\alpha$ | H | C | V | ARI | AMI | FMI | Q | NSD |
|---|---|---|---|---|---|---|---|---|
| 0 | 0.23 | **0.29** | 0.25 | 0.05 | 0.25 | **0.26** | 0.25 | 0.49 |
| 0.1 | **0.26** | **0.29** | **0.28** | **0.10** | **0.27** | **0.26** | 0.29 | 0.61 |
| 1 | 0.23 | 0.24 | 0.23 | 0.04 | 0.23 | 0.20 | **0.30** | **0.65** |
| 10 | 0.19 | 0.22 | 0.20 | 0.00 | 0.19 | 0.20 | 0.23 | 0.53 |

Table 3: Impact of noise on clustering performance (WS dataset).

$\alpha = 0$

| noise | H | C | V | ARI | AMI | FMI | Q | std |
|---|---|---|---|---|---|---|---|---|
| 0 % | 0.23 | 0.29 | 0.25 | 0.05 | 0.25 | 0.26 | 0.25 | 0.49 |
| 1 % | 0.00 | 0.49 | 0.00 | 0.00 | 0.00 | 0.39 | 0.00 | 0 |
| 5 % | 0.00 | 0.49 | 0.00 | 0.00 | 0.00 | 0.39 | 0.00 | 0 |
| 10 % | 0.00 | 0.49 | 0.00 | 0.00 | 0.00 | 0.39 | 0.00 | 0 |

$\alpha = 1$

| noise | H | C | V | ARI | AMI | FMI | Q | std |
|---|---|---|---|---|---|---|---|---|
| 0 % | 0.23 | 0.24 | 0.23 | 0.04 | 0.23 | 0.2 | 0.3 | 0.65 |
| 1 % | 0.24 | 0.24 | 0.24 | 0.04 | 0.23 | 0.2 | 0.3 | 0.66 |
| 5 % | 0.23 | 0.23 | 0.23 | 0.05 | 0.22 | 0.2 | 0.3 | 0.67 |
| 10 % | 0.24 | 0.23 | 0.23 | 0.05 | 0.23 | 0.2 | 0.3 | 0.67 |

## 6   CONCLUSION AND PERSPECTIVES

In this paper, we have provided a simple explanation for the well-known benefits of regularization on spectral embedding. Specifically, regularization forces the embedding to focus on the largest clusters, making the embedding more robust to noise. This result was obtained through the explicit characterization of the embedding for a simple block model, and extended to bipartite graphs.

An interesting perspective of our work is the extension to *stochastic* block models, using for instance the concentration results proved in (Lei et al., 2015; Le et al., 2017). Another problem of interest is the impact of regularization on other downstream tasks, like link prediction. Finally, we would like to further explore the impact of the regularization parameter, exploiting the theoretical results presented in this paper.

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

## APPENDIX

We provide of proof of Theorem 2 as well as a complete set of experimental results.

## A    REGULARIZATION OF BIPARTITE GRAPHS

The proof of Theorem 2 follows the same workflow as that of Theorem 1. Let $Z_1 \in \mathbb{R}^{n \times K}$ and $Z_2 \in \mathbb{R}^{m \times K}$ be the left and right membership matrices for the block matrix $B \in \mathbb{R}^{n \times m}$. The aggregated matrix is $\bar{B} = Z_1^T B Z_2 \in \mathbb{R}^{K \times K}$. The diagonal matrices of block sizes are $W_1 = Z_1^T Z_1$ and $W_2 = Z_2^T Z_2$. We have the equivalent of Proposition 1:

**Proposition 2.** *Let $x_1, x_2$ be a solution to the generalized singular value problem:*

$$\begin{cases} Bx_2 = \sigma D_1 x_1 \\ B^T x_1 = \sigma D_2 x_2 \end{cases}$$

*Then either $Z_1^T x_1 = Z_2^T x_2 = 0$ and $\sigma = 0$ or $x_1 = Z_1 y_1$ and $x_2 = Z_2 y_2$ where $y_1, y_2$ is a solution to the generalized singular value problem:*

$$\begin{cases} \bar{B} y_2 = \sigma \bar{D}_1 y_1, \\ \bar{B}^T y_1 = \sigma \bar{D}_2 y_2. \end{cases}$$

*Proof.* Since the rank of $B$ is equal to $K$, there are $n-K$ pairs of singular vectors $(x_1, x_2)$ associated with the singular values 0, each satisfying $Z_1^T x_1 = 0$ and $Z_2^T x_2 = 0$. By orthogonality, the other pairs of singular vectors satisfy $x_1 = Z_1 y_1$ and $x_2 = Z_2 y_2$ for some vectors $y_1, y_2 \in \mathbb{R}^K$. By replacing these in the original generalized singular value problem, we get that $(y_1, y_2)$ is a solution to the generalized singular value problem for the aggregate graph. $\square$

In the following, we focus on the block model described in Section 4, where $B = Z_1 Z_2^T$.

*Proof of Lemma 4.* The generalized eigenvalue problem (6) associated with the regularized matrix $A_\alpha$ is equivalent to the generalized SVD of the regularized biadjacency matrix $B_\alpha$:

$$\begin{cases} B_\alpha x_2 = \sigma D_{\alpha,1} x_1 \\ B_\alpha^T x_1 = \sigma D_{\alpha,2} x_2, \end{cases}$$

with $\sigma = 1 - \lambda$.

In view of Proposition 2, the singular value $\sigma = 0$ has multiplicity $n - K$, meaning that the eigenvalue $\lambda = 1$ has multiplicity $n - K$. Since the graph is connected, the eigenvalue 0 has multiplicity 1. The proof then follows from the observation that if $(x_1, x_2)$ is a pair of singular vectors for the singular value $\sigma$, then the vectors $x = (x_1, \pm x_2)^T$ are eigenvectors for the eigenvalues $1 - \sigma, 1 + \sigma$.

*Proof of Lemma 5.* By Proposition 2, we can focus on the generalized singular value problem for the aggregate graph:

$$\begin{cases} \bar{B}_\alpha y_2 = \sigma \bar{D}_{\alpha,1} y_1 \\ \bar{B}_\alpha^T y_1 = \sigma \bar{D}_{\alpha,2} y_2, \end{cases}$$

Since

$$\bar{B}_\alpha = W_1 (I_K + \alpha J_K) W_2,$$

and

$$\begin{cases} \bar{D}_{\alpha,1} = W_1(W_2 + \alpha nI), \\ \bar{D}_{\alpha,2} = W_2(W_1 + \alpha mI), \end{cases}$$

we have:

$$\begin{cases} W_1(I_K + \alpha J_K)W_2 y_2 = W_1(W_2 + \alpha nI)y_1\sigma, \\ W_2(I_K + \alpha J_K)W_1 y_1 = W_2(W_1 + \alpha mI)y_2\sigma. \end{cases}$$

Observing that $J_K W_1 y_1 \propto 1_K$ and $J_K W_2 y_2 \propto 1_K$, we get:

$$\begin{cases} (W_2 + \alpha mI_K)y_1\sigma - W_2 y_2 \propto 1_K, \\ (W_1 + \alpha nI_K)y_2\sigma - W_1 y_1 \propto 1_K. \end{cases}$$

As two diagonal matrices commute, we obtain:

$$\begin{cases} (W_1 + \alpha nI_K)(W_2 + \alpha mI_K)y_1\sigma - W_1 W_2 y_1 = \big(\eta_1(W_1 + \alpha nI_K) + \eta_2 W_2\big)1_K, \\ (W_1 + \alpha nI_K)(W_2 + \alpha mI_K)y_2\sigma - W_1 W_2 y_2 = \big(\eta_1 W_1 + \eta_2(W_2 + \alpha mI_K)\big)1_K, \end{cases}$$

for some constants $\eta_1, \eta_2$, and

$$\begin{cases} y_{1,j} = \dfrac{\eta_1(n_j + \alpha n) + \eta_2 m_j}{(n_j + \alpha n)(m_j + \alpha m)\sigma - n_j m_j}, \\ y_{2,j} = \dfrac{\eta_1 n_j + \eta_2(m_j + \alpha m)}{(n_j + \alpha n)(m_j + \alpha m)\sigma - n_j m_j}. \end{cases}$$

Letting $s_1 = -\text{sign}(\eta_1(n_j + \alpha n) + \eta_2 m_j)$ and $s_2 = -\text{sign}(\eta_1 n_j + \eta_2(m_j + \alpha m))$, we get:

$$\text{sign}(y_{1,j}) = s_1 \iff \text{sign}(y_{2,j}) = s_2 \iff \frac{n_j m_j}{(n_j + \alpha n)(m_j + \alpha m)} \geq \sigma = 1 - \lambda,$$

and the result follows from the fact that $x_1 = Z_1 y_1$ and $x_2 = Z_2 y_2$.

*Proof of Lemma 6.* The proof is the same as that of Lemma 3, where the threshold values follow from Lemma 5:

$$\mu_j = 1 - \frac{n_j m_j}{(n_j + \alpha n)(m_j + \alpha m)}.$$

*Proof of Theorem 2.* Let $x$ be the $j$-th column of the matrix $X$, for some $j \in \{2, \ldots, k\}$. In view of Lemma 6, this is the eigenvector associated with eigenvalue $\lambda_j \in (\mu_{j-1}, \mu_j)$. In view of Lemma 4, all entries of $x$ corresponding to blocks of size $n_1, n_2 \ldots, n_{j-1}$ have the same sign, the other having the opposite sign.

## B  EXPERIMENTAL RESULTS

In this section, we present more extensive experimental results.

Tables 4 and 5 present results for the same experiment as in Table 2 but for different values of $K$, namely $K = 2$ (bisection of the graph) and $K = K_{\text{truth}}/2$ (half of the ground-truth value). As for $K = K_{\text{true}}$, regularization generally improves clustering performance. However, the optimal value of $\alpha$ remains both dataset dependent and metric dependent. Note that, for the NG and WS datasets, the clustering remains trivial in the case $K = 2$, one cluster containing all the nodes, until a certain amount of regularization.

Table 6 presents the different scores for both types of regularization on the NG dataset. As we can see, preserving the bipartite structure of the graph leads to slightly better performance.

Finally, Table 7 shows the impact of regularization in the presence of noise for the NG dataset. The conclusions are similar as for the WS dataset: regularization makes the spectral embedding much more robust to noise.

Table 4: Impact of regularization on clustering performance. $K = 2$.

SBM

| $\alpha$ | H | C | V | ARI | AMI | FMI | Q | NSD |
|---|---|---|---|---|---|---|---|---|
| 0 | 0.00 | 0.43 | 0.00 | 0.0 | 0.0 | **0.10** | 0.00 | 0.01 |
| 0.1 | 0.00 | **0.47** | 0.00 | 0.0 | 0.0 | **0.10** | 0.00 | 0.00 |
| 1 | **0.01** | 0.04 | **0.01** | 0.0 | 0.0 | 0.07 | **0.34** | **0.83** |
| 10 | **0.01** | 0.09 | **0.01** | 0.0 | 0.0 | 0.09 | 0.13 | 0.22 |

NG

| $\alpha$ | H | C | V | ARI | AMI | FMI | Q | NSD |
|---|---|---|---|---|---|---|---|---|
| 0 | 0.00 | 0.36 | 0.00 | 0.00 | 0.00 | 0.23 | 0.00 | 0.00 |
| 0.1 | 0.00 | 0.36 | 0.00 | 0.00 | 0.00 | 0.23 | 0.00 | 0.00 |
| 1 | **0.15** | **0.72** | **0.25** | **0.06** | **0.25** | **0.28** | **0.16** | **0.63** |
| 10 | 0.12 | 0.61 | 0.20 | 0.04 | 0.20 | 0.26 | 0.13 | 0.51 |

WS

| $\alpha$ | H | C | V | ARI | AMI | FMI | Q | NSD |
|---|---|---|---|---|---|---|---|---|
| 0 | 0.00 | **0.49** | 0.00 | 0.00 | 0.00 | **0.39** | 0.00 | 0.00 |
| 0.1 | **0.07** | 0.42 | **0.13** | 0.00 | **0.12** | 0.34 | 0.09 | **0.26** |
| 1 | 0.03 | 0.27 | 0.05 | -0.01 | 0.05 | 0.35 | 0.09 | 0.13 |
| 10 | 0.02 | 0.16 | 0.04 | -0.02 | 0.03 | 0.34 | **0.10** | 0.16 |

Table 5: Impact of regularization on clustering performance. $K = K_{\text{true}}/2$.

SBM

| $\alpha$ | H | C | V | ARI | AMI | FMI | Q | NSD |
|---|---|---|---|---|---|---|---|---|
| 0 | 0.08 | 0.21 | 0.11 | 0.0 | 0.00 | **0.05** | 0.41 | 0.47 |
| 0.1 | 0.20 | 0.27 | 0.23 | 0.0 | **0.01** | 0.02 | **0.55** | 0.84 |
| 1 | **0.24** | **0.29** | **0.26** | 0.0 | 0.00 | 0.02 | 0.54 | **0.90** |
| 10 | 0.19 | 0.28 | 0.23 | 0.0 | 0.00 | 0.03 | 0.40 | 0.70 |

NG

| $\alpha$ | H | C | V | ARI | AMI | FMI | Q | NSD |
|---|---|---|---|---|---|---|---|---|
| 0 | 0.27 | **0.76** | 0.40 | 0.11 | 0.39 | 0.31 | 0.20 | 0.41 |
| 0.1 | 0.28 | 0.73 | 0.41 | 0.11 | 0.40 | 0.30 | 0.18 | 0.43 |
| 1 | **0.38** | 0.72 | **0.50** | **0.18** | **0.50** | **0.34** | **0.21** | **0.57** |
| 10 | 0.31 | 0.62 | 0.42 | 0.11 | 0.42 | 0.27 | 0.17 | 0.51 |

WS

| $\alpha$ | H | C | V | ARI | AMI | FMI | Q | NSD |
|---|---|---|---|---|---|---|---|---|
| 0 | 0.23 | **0.29** | 0.25 | 0.05 | 0.25 | **0.26** | 0.25 | 0.49 |
| 0.1 | **0.26** | **0.29** | **0.28** | **0.10** | **0.27** | **0.26** | 0.29 | 0.61 |
| 1 | 0.23 | 0.24 | 0.23 | 0.04 | 0.23 | 0.20 | **0.30** | **0.65** |
| 10 | 0.19 | 0.22 | 0.20 | -0.00 | 0.19 | 0.20 | 0.23 | 0.53 |

Table 6: Regularization of the adjacency vs. biadjacency matrix on the NG dataset ($\alpha = 1$).

$K = K_{\text{true}}/2$

|        | H    | C    | V    | ARI  | AMI  | FMI  | Q    | std  |
|--------|------|------|------|------|------|------|------|------|
| Adj.   | 0.38 | 0.72 | 0.50 | 0.18 | 0.50 | 0.34 | 0.21 | 0.57 |
| Biadj. | **0.41** | 0.72 | **0.52** | **0.19** | **0.52** | **0.35** | 0.21 | **0.61** |

$K = K_{\text{true}}$

|        | H    | C    | V    | ARI  | AMI  | FMI  | Q    | std  |
|--------|------|------|------|------|------|------|------|------|
| Adj.   | 0.46 | 0.67 | 0.54 | 0.20 | 0.54 | 0.33 | 0.2  | 0.60 |
| Biadj. | **0.47** | **0.68** | **0.56** | **0.21** | **0.55** | **0.34** | 0.2  | **0.61** |

Table 7: Impact of noise on clustering performance (NG dataset).

$\alpha = 0$

| noise | H    | C    | V    | ARI  | AMI  | FMI  | Q    | std  |
|-------|------|------|------|------|------|------|------|------|
| 0 %   | 0.40 | 0.70 | 0.51 | 0.19 | 0.50 | 0.34 | 0.21 | 0.55 |
| 1 %   | 0.00 | 1.00 | 0.00 | 0.00 | 0.00 | 0.23 | 0.00 | 0    |
| 5 %   | 0.14 | 0.65 | 0.23 | 0.06 | 0.23 | 0.27 | 0.13 | 0.30 |
| 10 %  | 0.00 | 0.36 | 0.01 | 0.00 | 0.00 | 0.23 | 0.00 | 0.00 |

$\alpha = 1$

| noise | H    | C    | V    | ARI  | AMI  | FMI  | Q    | std  |
|-------|------|------|------|------|------|------|------|------|
| 0 %   | 0.46 | 0.67 | 0.54 | 0.20 | 0.54 | 0.33 | 0.2  | 0.60 |
| 1 %   | 0.48 | 0.66 | 0.56 | 0.21 | 0.56 | 0.33 | 0.2  | 0.64 |
| 5 %   | 0.49 | 0.66 | 0.56 | 0.23 | 0.56 | 0.34 | 0.2  | 0.66 |
| 10 %  | 0.45 | 0.66 | 0.54 | 0.20 | 0.54 | 0.33 | 0.2  | 0.59 |

