# OpenReview forum: "Spectral  Embedding of Regularized Block Models"
_ICLR.cc/2020/Conference — Accept (Spotlight)_

### Official Review · AnonReviewer2 · 2019-10-24
**Official Blind Review #2**

**Rating:** 6

**Review:**

The paper explains through a block model the impact of the complete graph regularization, intended as adding to all the entries of the adjacency matrix a constant. The paper is a nice balance between theory and practical effect, since it shows that at the end spectral embedding has an impact on larger connected block units of the graph, discarding isolated nodes.

It also introduces the problem in a gentle way, so that the range of possible readers is wide.

In general I'm happy with the paper, no major lacks on my side

Suggestions: it is not clear how to get to Eq.7), the authors should explain the last passage before that equation a little more?  How the values of the noise alpha have been selected? How the approach scales with the number of blocks, in term of complexity?



**Experience Assessment:**

I have read many papers in this area.

**Review Assessment: Checking Correctness Of Derivations And Theory:**

I assessed the sensibility of the derivations and theory.

**Review Assessment: Checking Correctness Of Experiments:**

I assessed the sensibility of the experiments.

**Review Assessment: Thoroughness In Paper Reading:**

I read the paper at least twice and used my best judgement in assessing the paper.

---

> ### Author Response · Authors · 2019-11-06
> **Response to reviewer 2**
>
> Thanks for your comments and suggestions.
>
> * We have detailed the derivation of Eq (7) (see the revised version).
> * Selecting good values for alpha is an interesting question, that is indeed not addressed in our paper. We simply recommend to use the relative value of alpha with respect to the total weight of the graph, as in our experiments. We have selected a representative range of magnitudes (0, 0.1, 1, 10) to illustrate the sensitivity of the results to this relative parameter.
> * Our main result (Theorem 1) gives the structure of the spectral embedding, independently of the number of blocks. The computation of the spectral embedding of the block model requires to solve an eigenvalue problem in dimension K (the number of blocks), whose complexity depends on the chosen solver.

---

### Official Review · AnonReviewer1 · 2019-10-27
**Official Blind Review #1**

**Rating:** 8

**Review:**

This paper analyzes the effect of regularization on spectral embeddings in a deterministic block model and explicitly characterizes the spectra of the Laplacian of the regularized graph in terms of the regularization parameter and block sizes. To my knowledge, this has not been done before. Prior work either derives sufficient conditions for the recovery of all blocks in the asymptotic limit of an infinite number of nodes in the case of (Joseph & Yu, 2016), or lower bounds the number of small eigenvalues of the Laplacian of the unregularized graph on random graphs in expectation (therefore arguing in favor of regularization) in the case of (Zhang & Rohe, 2018). This paper, on the other hand, gives a precise characterization of the eigenvalues and eigenvectors (albeit in the case of a deterministic graph); the results are elegant and the analysis uses simple elementary techniques, which is very satisfying and seems to be easy to build on. The authors mention that they would like to extend this analysis to stochastic block models, which would indeed be interesting. The paper is also well written and the results are clearly presented. Overall, this is a nice contribution to spectral graph theory and so I recommend acceptance.

**Experience Assessment:**

I have read many papers in this area.

**Review Assessment: Checking Correctness Of Derivations And Theory:**

I assessed the sensibility of the derivations and theory.

**Review Assessment: Checking Correctness Of Experiments:**

I assessed the sensibility of the experiments.

**Review Assessment: Thoroughness In Paper Reading:**

I read the paper thoroughly.

---

### Decision · Program_Chairs · 2019-12-19

**Decision:**

Accept (Spotlight)

**Comment:**

The paper proposes a nice and easy way to regularize spectral graph embeddings, and explains the effect through a nice set of experiments. Therefore, I recommend acceptance.